# Position: The Categorization of Race in ML is a Flawed Premise

**Miriam Doh** [1,2]  **Benedikt Höltgen** [3]  **Piera Riccio** [4]  **Nuria Oliver** [4]

## Abstract

This position paper critiques the reliance on rigid racial taxonomies in machine learning, exposing their U.S.-centric nature and lack of global applicability—particularly in Europe, where race categories are not commonly used. These classifications oversimplify racial identity, erasing the experiences of mixed-race individuals and reinforcing outdated essentialist views that contradict the social construction of race. We suggest research agendas in machine learning that move beyond categorical variables to better address discrimination and social inequality.

## 1. Introduction

"A race is only a sort of average of a large number of individuals; and averages differ from one another much less than individuals. Popular impression exaggerates the differences, accurate measurements reduce them" (Kroeber, 1948)

The concept of separate human races arose in the 17th and 18th centuries and was used by Westerners to justify slavery despite their Christian faith (Smedley & Smedley, 2005). This notion persisted into the 20th century and was closely tied to early statistics and eugenics, with pioneers like Galton, Pearson, and Fisher reinforcing the idea of biological racial separability. However, anthropologists have increasingly contested these racial distinctions, recognizing them as social constructs rather than biologically discrete entities (Kroeber, 1948). The "one-drop rule" in the U.S. and the Apartheid regime's classification system in South Africa illustrate how racial categories have been historically fluid and politically motivated (Bowker & Star, 1999).

For decades, scholars across disciplines have emphasized that racial classifications are neither genetically discrete

(Wilson et al., 2001) nor can they be reliably measured or considered scientifically meaningful (Smedley & Smedley, 2005). Instead, race is understood as a construct whose meaning is constituted by social arrangements, practices, and intersubjective beliefs (Hu & Kohler-Hausmann, 2020). This understanding implies that there is no "correct" racial taxonomy derived from biology, as perceptions of race depend on both phenotypic traits and contextual interpretations.

Despite this knowledge, the Machine Learning (ML) research community has often relied on datasets that label race as a categorical variable, treating it as a "ground-truth" that simplifies its social and historical complexity (Abdu et al., 2023). Racial labels such as "White," "Black," or "Asian" are indeed widely used in image (Phillips et al., 2000; Ricanek & Tesafaye, 2006; Zhang et al., 2017; Karkkainen & Joo, 2021) and tabular (Kohavi et al., 1996; Angwin et al., 2016) datasets, often adopting U.S. census-based racial classifications without contextualization, reinforcing North American constructs in global ML research.

By using these categorizations, ML models fail to account for the social, cultural and historical contexts that shape racial identities. Even when these labels are well-intended and used for fairness audits (Wang & Deng, 2019; Ravishankar et al., 2023; Yucer et al., 2024b), they reflect outdated racial theories. This lack of critique has real-world consequences: from biased hiring algorithms to unequal healthcare diagnostics, these systems risk exacerbating disparities rather than addressing them. To move forward, we contend that the ML community should examine how racial data is collected, labeled and used, avoiding the instrumentalization of ethical concepts, to the point that "bias becomes a form of numerical error to be corrected with better datasets, and ethics becomes a bureaucratic checklist to be inserted into the production flowchart" (Hong, 2023).

In this paper, we discuss the challenges and ethical concerns associated with the use of categorical race labels in ML systems. Thus, discussing the benefits for individuals to freely define their identities through racial categories (Rich, 2013; Stock et al., 2018) is out of its scope. Instead, we critically analyze ML practices, datasets, and the socio-historical construction of race (Section 2), emphasizing the dominance of U.S.-centric frameworks and their limitations

[1] ISIA Lab, Université de Mons, Mons, Belgium [2] IRIDIA Lab, Université Libre de Bruxelles, Ixelles, Belgium [3] University of Tübingen, Germany [4] Ellis Alicante, Spain. Correspondence to: Miriam Doh <miriam.doh@umons.ac.be>.

*Proceedings of the 42ⁿᵈ International Conference on Machine Learning*, Vancouver, Canada. PMLR 267, 2025. Copyright 2025 by the author(s).

from the perspective of Europe (Section 3) and the "mixed race" community (Section 4). In addition, we address the broader issue of racial reification and stereotyping in ML (Section 5). Finally, we propose a research agenda that avoids categorical race labels by integrating context and domain knowledge to more effectively tackle discrimination and social inequality (Section 6).

**In summary, we argue that racial categories should be abandoned in ML research whenever possible, as they reinforce outdated essentialist views and fail to account for the complexity of human identities. We do not argue that racial discrimination should be ignored—on the contrary, we contend that fairness frameworks in machine learning must be reimagined in ways that effectively address bias without reinforcing the very inequalities they aim to mitigate.**

## 2. Related Work

Several studies have criticized the use of fixed racial categories in AI from different perspectives. First, the process of defining racial taxonomies has been questioned, calling for more careful and contextual choices, justification, and documentation (Crawford & Paglen, 2021; Gebru et al., 2021; Khan & Fu, 2021; Abdu et al., 2023; Mickel, 2024). Second, the use of racial categories in assessing fairness and discrimination in ML has been challenged, calling for the use of different categories (Benthall & Haynes, 2019; Hanna et al., 2020; Belitz et al., 2023; Jaime & Kern, 2024). In contrast, we argue that methods aimed at measuring discrimination should go beyond using a categorical race variable.

Recent work has presented alternatives to categorical race labels both in vision and tabular data. In computer vision, the use of visual cues instead of race has been proposed (Buolamwini & Gebru, 2018), almost exclusively focusing on skin tone (Ajmal et al., 2021; Bloomberg, 2023; Groh et al., 2022; Currie et al., 2024). We argue that other phenotypic features should be considered while avoiding "racial phenotypes" (Yucer et al., 2022; 2024a), which run the risk of reifying race along phenotypic lines (Hanna et al., 2020; Crawford & Paglen, 2021). In the case of tabular data, the most closely related work to ours concerns subgroup fairness (Kearns et al., 2018) and multi-calibration (Hébert-Johnson et al., 2018). However, the former approach still relies on predefined race categories while the latter solely aims at general predictive quality without incorporating context or distinguishing between features.

## 3. Race taxonomies in Machine Learning: US centrism vs the European perspective

Race is often invoked as a protected attribute in the context of algorithmic fairness in ML, building on U.S. discrimina-

tion doctrines of disparate treatment or impact (Barocas & Selbst, 2016). This area has been shaped by seminal works, such as the 2016 ProPublica analysis of the COMPAS recidivism prediction algorithm, reporting higher error rates for black than for white inmates (Angwin et al., 2016). For the past decade, numerous authors have argued for fairness metrics defined in terms of comparisons between socially relevant groups, such as equal error rates or equal outcomes, and ways to optimize for them (Barocas et al., 2023). In these works, race is typically considered a categorical attribute as any other, often adopting the taxonomy of the U.S. census, as we discuss below. Race has also been extensively used in computer vision, where many datasets with human faces contain race labels that are used to compare error rates across different racial groups (Garcia et al., 2019), provide a measure of dataset diversity (Chen, 2020), or as target variables for models serving a variety of purposes, including "security and defense, surveillance, human computer interface (HCI), biometric-based identification" (Fu et al., 2014).

Recent studies highlight how fairness research continues to rely on predefined racial classifications without critical engagement. Abdu et al. analyzed 60 ACM FAccT (2018-2020) papers (Abdu et al., 2023), showing that most use racial categories without justification, primarily adopting labels from pre-existing datasets. Extending this analysis, we reviewed 78 ICML and CVPR papers from 2023 and 2024, selecting those with "fair" in the title for ICML and "fair"/"bias" for CVPR, as the latter yielded too few results when searching only for "fair".

Among these 78 papers, 45 explicitly discussed group fairness. Within this subset, 29 relied on racial categories as a protected attribute, all adopting rigid racial classifications inherited from existing datasets—confirming the persistence of these taxonomies in fairness research (Abdu et al., 2023). This reliance suggests a continued adoption of established classification schemes without a critical reassessment of their validity. To further investigate this trend, we examined the most widely used ML datasets based on citation counts from Google Scholar and their recurrence in our paper analysis. Our findings confirm that these datasets overwhelmingly adopt racial classifications derived from U.S. census categories (Appendix B), a pattern also highlighted in (Abdu et al., 2023). Even when explicit census labels are absent, the underlying racial categories often reflect U.S.-centric taxonomies, reinforcing rigid classification structures. Notably, none of these datasets provide critical reflection of their contextual meaning or applicability beyond the United States (Benthall & Haynes, 2019; Khan & Fu, 2021; Mickel, 2024) (see Appendix C).

Despite the cultural vicinity, the legal and institutional framework in Europe contrasts sharply with the U.S. ap-

proach, where racial categories are routinely collected and integrated into governance, research, and AI datasets. Mainly as a result of post-World War II efforts to combat racialization and discrimination, race is widely rejected as a legitimate classification category in Europe (Simon, 2017; Osanami Törngren, 2020; Rodríguez-García et al., 2021). Also at the legislative level, the European Union has institutionalized its avoidance of racial classification. For instance, the General Data Protection Regulation (GDPR) (Union, 2016) restricts the collection of racial or ethnic data, permitting it only under narrowly defined conditions, such as explicit consent or when serving a significant public interest. In practice, many European states remain hesitant to engage with race as an analytical category, relying instead on indirect indicators, such as nationality, language spoken at home, or parental country of birth (Westin, 2003; Simon, 2015; European Commission, 2017; Bereni et al., 2021; Osanami Törngren et al., 2021). Indeed, the Court of Justice of the European Union ruled that 'ethnic origin cannot be determined on the basis of a single criterion' and judicial decisions on racial or ethnic discrimination need to consider multiple factors (CJEU, 2017). Furthermore, different member states have different standards on which attributes are relevant for racial discrimination (Commission et al., 2024), highlighting the importance of local context.

As AI systems become increasingly regulated under frameworks such as the EU AI Act (Commission, 2021), with explicit requirements regarding algorithmic fairness and non-discrimination, the direct adoption of U.S.-based frameworks raises significant questions (Wachter et al., 2021; Engler, 2023). However, even if the institutional approach to dealing with racial categories is more permissive in the United States, criticism of such taxonomies has not been exclusive to Europe. Even in the U.S., racial beliefs have long been said to "constitute myths about the diversity in the human species and about the abilities and behavior of people homogenized into "racial" categories" (American Anthropological Association, 1998).

In line with this, we argue that racial categories should be abandoned in ML research whenever possible, incorporating context and domain knowledge instead to better address discrimination. In the following sections, we support this position by identifying two problems which highlight why racial categorization in ML can be both conceptually flawed and harmful. We start by demonstrating that ML frameworks fail to represent mixed-race identities and enforce reductive classifications.

## 4. The Mixed-race Problem in ML

Mixed-race individuals, defined as those with parents from different racial backgrounds or who belong to multiple racial groups (Oxford English Dictionary, 2025b),

embody the complexity and fluidity of racial identities. These individuals navigate multiple cultural and racial contexts, resisting singular classification and exposing the limitations of existing taxonomies (Remedios & Chasteen, 2013; Camara, 2016; McWatt, 2020). Their dual positioning—simultaneously within and beyond established categories—makes them emblematic of identities that defy categorical frameworks (Ali, 2020). At the heart of this experience lies the concept of "in-betweenness", which highlights both the richness of multiplicity and the challenges of fragmented affiliations (Floro, 2018; Brocket, 2020). This liminal space often amplifies marginalization, reflecting how societal and institutional structures struggle to accommodate fluid identities (Camara, 2016; Ali, 2020; Nilipour, 2021).

A fundamental question emerges: *how should mixed-race identities be classified within categorical race taxonomies in AI?* There are arguably five possible ways to handle mixed-race categories in categorical race taxonomies, which we will discuss as approaches (A) to (E) below, with (A) to (C) being used in practice (Abdu et al., 2023).

**Approach (A)** assumes that each person only belongs to one race, hence ignoring the reality of mixed-race individuals and invalidating their identity by forcing them to claim only one aspect of it (Miles, 2020; Ford et al., 2021). This approach is largely driven by two factors: (1) the preference for mutually exclusive racial classification schemes, which streamline computational modeling at the expense of identity fluidity, and (2) the push for computational efficiency, where minimizing the number of racial categories simplifies both the analysis and fairness auditing (Abdu et al., 2023; Mickel, 2024). However, these practical considerations come at the cost of inclusivity and representational accuracy, ultimately reinforcing reductive racial frameworks rather than challenging them. This seems to be the most common approach in ML and especially in computer vision, as exemplified by the `FairFace` (Karkkainen & Joo, 2021) dataset. Marketed as a "fair" dataset, it was designed to mitigate the racial imbalances in existing face datasets by introducing seven racial groups. The authors explicitly frame their motivation as follows: *"Existing public face datasets are strongly biased toward Caucasian faces, and other races (e.g., Latino) are significantly underrepresented."* This illustrates that under the assumption of rigid racial taxonomies, even well-intended approaches can become highly exclusive.

**Approach (B)** includes a single "Mixed-race" category, as it is the case of the American Community Survey (ACS) Public Use Microdata Sample (PUMS) data (United States Census Bureau), from which the `folktables` dataset (Ding et al., 2021) is derived. This choice may seem to be an efficient solution given that, in the U.S. for instance, many mixed-race Black/White individuals identify as Black (and

are treated as such) since the early 20th century (Thompson, 2013), and hence the mixed-race category only comprised 2.9% of U.S. census respondents as of 2010. In 2020, however, this percentage increased to 10.2%, presumably "largely due to the improvements to the design [and processing]" of the survey (United States Census Bureau, 2021).[1] However, this approach is also highly reductive: Why should, for instance, "Asian-White" be grouped together with "Black-Hispanic"?

We illustrate this limitation with an example in computer vision, using embeddings extracted with CLIP ViT-B/32 (Radford et al., 2021) from the `Chicago Face Database` (`CFD`) and its extension `CFD-MR` (Ma et al., 2015; 2021). `CFD` consists of images of 597 unique individuals with their self-reported race (Asian, Black, Latino, and White), and `CFD-MR` includes images of 88 unique individuals who self-reported multi-racial ancestry (Mixed-Race). We conduct an experiment comparing the images of faces in the Asian, Black, White and Mixed-Race groups, ensuring balance across gender and sample size. As shown in Figure 1, we observe significant overlap between Mixed-Race samples (shown as magenta dots in the right-hand side of the Figure) and other racial groups. This blurring of the boundaries between groups in the case of mixed-race individuals further illustrates the limitation of relying on rigid racial taxonomies.

We acknowledge that CFD-MR's small sample size limits its statistical robustness; however, to our knowledge, it is the only publicly available dataset including self-reported mixed-race identity, enabling analysis of mixed-race complexity without imposing external labels. The scarcity of larger, self-reported mixed-race datasets highlights a gap in available data and a shortfall in current collection practices—underscoring the urgent need for more nuanced, self-defined datasets.

**Approach (C)** subsumes mixed-race under "Other", as exemplified by (Xian et al., 2023). This option not only inherits the limitations of using a single "Mixed-Race" category but even exacerbates them by adding to all possible variations of mixed-race individuals any other individuals that do not fit into the imposed categories.

In addition to these three approaches, we describe below two potential approaches that have not been pursued yet in the ML literature.

**Approach (D)** accounts for all mixed categories separately, *e.g.*, "Black/White", "Black/Hispanic", etc. However, this approach does not seem advisable for two reasons. First, it would lead to a combinatorial explosion, as for $k$ categories,

---

[1]This dependence on framing as well as the fact that indicating multiple races was only possible after long negotiations in the 90s (Robbin, 2000) exemplifies the volatility of race taxonomies.

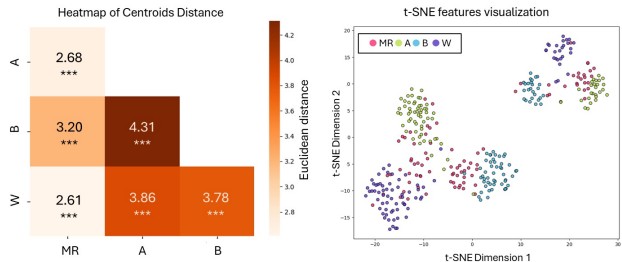

*Figure 1.* Embedding visualization for four racial groups from the Chicago Face Dataset and its extension, CFD-MR: Mixed-Race (MR), Asian (A), Black (B), and White (W). Left: heatmap of the Euclidean distances between group centroids, with significance: *** ($p < 0.001$), ** ($p < 0.01$), * ($p < 0.05$). Right: t-SNE plot of the embeddings of each individual image. Note how Mixed-Race individuals (depicted in magenta) occupy an intermediate space between other groups.

this would lead to $2^k - 1$ possible labels. Second, the categorical nature of the variable would mean that, for instance, "Black/Hispanic" would stand in the same relationship to "Black" as to "White", which is clearly sub-optimal.

**Approach (E)** proposes to allow individuals to have multiple racial labels instead of being forced into a single category. While this better reflects the reality of mixed-race identities, it is not used in practice due to both data availability issues and the technical complexity of multi-label analysis. For instance, while the U.S. Census initially records race data in this way, it is later simplified into single-label categories in the ACS PUMS database, and hence also `folktables` (approach (B)). Even if such data were available, implementing this approach in ML would require new methods to handle overlapping groups, which most fairness frameworks do not support.

Rather than attempting to develop new technical solutions for such categorical approaches, we argue that race labels should be avoided altogether. We provide further justification for this position in the following section.

## 5. From Reifying to Stereotyping Race in AI

The concept of reification refers to the process of transforming abstract ideas into concrete, seemingly natural entities (Cambridge Dictionary, 2025). In the context of AI, racial reification occurs when socially constructed racial categories are encoded as computer-readable attributes and, thus, appear to be measurable and biologically real. This process not only replicates racial classification systems but entrenches race within digital infrastructures. It thereby reinforces the false notion that racial groups are stable, objective, and biologically determined rather than historically

contingent and socially constructed (Monea, 2019; Hanna et al., 2020; Khan & Fu, 2021; Yucer et al., 2024b). These mechanisms, deeply embedded in machine learning systems, mirror historical physiognomic classification, where racial typologies were used to categorize, rank, and regulate human difference (Poole, 2005; Benjamin, 2019; Moss, 2021; Zhao et al., 2024).

A key issue in AI's reification of race lies in how image datasets obtain the labels for racial categories. Many widely used datasets, such as `FairFace` (Karkkainen & Joo, 2021), rely on manual annotation, where individuals classify faces into racial categories without any contextual information. These categorizations, often based on a subjective visual assessment, do not account for the fluidity of racial identity, regional variations, or self-identification, yet they serve as the foundation for how ML models "learn" to recognize or consider race (Khan & Fu, 2021). In addition, these datasets impose a singular, racialized gaze that fixes the representation of marginalized identities into stereotypical, externally imposed frameworks. A clear example of this can be found in `ImageNet` (Deng et al., 2009), where the category "Black person" was represented in part by images of individuals performing *blackface*, illustrating how racialized bodies have historically been captured through a *white gaze* that dissects, categorizes, and fixes them in structures of power, rather than allowing for self-representation (Monea, 2019; Crawford & Paglen, 2021).

While ML-based classification systems reinforce racial categories on a large scale, generative systems for images, videos, and even audio push this further by enabling the creation of content tied to racial classifications (Bianchi et al., 2023; Luccioni et al., 2024; Ghosh et al., 2024; Zhou et al., 2024). Unlike static datasets which have a finite number of datapoints, Generative AI models can synthesize entirely new datasets, reinforcing and expanding racial stereotypes with no human verification or historical grounding.

To illustrate this phenomenon, we conducted an experiment with Stable Diffusion 3.5 Large (Huggingface, 2024), Midjourney (MidJourney, 2024), and DALL·E 3 (OpenAI, 2024), using the prompts "a mixed-race person" (MR-PROMPT) and "a mulatto[2] person" (MU-PROMPT). The inclusion of the term mulatto in this study was intentional, as it historically frames mixed-race identity within a rigid "Black"/"White" binary (Oxford English Dictionary, 2025a). We did not include prompts such as "Asian-White," which lack an equally established cultural reference and could introduce arbitrary assumptions about how these identities should appear.

---

[2]Disclaimer: The term *mulatto* is considered offensive in contemporary usage. It is used here solely for historical and analytical purposes to critically examine biases in Generative AI representations.

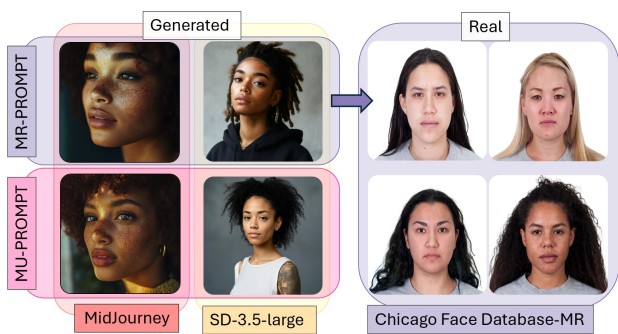

*Figure 2.* Comparison of AI-generated and real-world images of mixed-race individuals. The generated images come from two AI models, MidJourney and Stable Diffusion 3.5 Large, using different prompts: *"A mulatto person"* (MU-PROMPT) and *"a mixed-race person"* (MR-PROMPT). The real-world images are sourced from the Chicago Face Database-MR, showcasing greater phenotypic diversity.

Because Generative AI systems are trained on datasets reflecting historical and cultural biases, we hypothesized that they might replicate or reinforce such reductive categorizations. Consequently, while our conclusions are necessarily limited to the specific historical stereotypes associated with these terms, this focus ensures that our findings about the replication of pre-existing biases remain clearly interpretable.

To empirically validate this hypothesis, we then generated images using each model under both prompts. Specifically, 30 images per prompt were generated for both Stable Diffusion and Midjourney. Note that we were unable to create images with DALL·E 3 due to prompt moderation on the term "mulatto". Interestingly, both models produced nearly identical images across these prompts: individuals with medium-brown skin, curly or afro-textured hair, and phenotypic traits associated with a "Black"/"White" racial mix. Figure 2, left, shows an example of the generated images for each prompt and model. From a quantitative perspective, the cosine similarity between the embeddings (extracted using CLIP ViT-B/32) of the two sets of images was 0.8190 for Stable Diffusion and 0.7687 for MidJourney. This suggests that both models interpreted the MR-PROMPT and MU-PROMPT similarly, thus creating similar images for both prompts. Conversely, natural images exhibit a much wider range of phenotypic diversity than the images generated with AI models. For example, the images in the `CFD-MR` (Ma et al., 2021) showcase greater diversity under the "Mixed Race" category, as illustrated in Figure 2. From a quantitative perspective, the mean cosine similarity of embeddings (also extracted using CLIP ViT-B/32) within the `CFD-MR` dataset is 0.7883, whereas the generated images with the MR-PROMPT exhibit larger internal similarity

(0.8694 for Stable Diffusion and 0.8265 for Midjourney). Despite the small sample size, our analysis of generative AI outputs provides key insights into how these models replicate and reinforce existing racial stereotypes. More details on dataset composition and comparative analysis can be found in Appendix D."

Given the apparent limitations of existing frameworks to address race in ML, we present next an alternative approach that aims to more effectively account for racial dynamics and mitigate biases in ML systems.

## 6. Moving Beyond Race Labels in ML

In response to the concerns raised in this paper and more generally the constructivist understanding of race (Smedley & Smedley, 2005), we propose that ML research should abandon racial taxonomies where possible and instead focus on developing alternative methods that address identity-based disparities without reifying race as a biological category.

The challenge to tackle racial discrimination without fixed racial categories has been addressed in multiple disciplines. There is a growing shift towards fine-grained, context-sensitive approaches that challenge rigid racial classifications in a variety of fields, including biology (Yudell et al., 2016), economics (Rose, 2023), public health (Braveman & Parker Dominguez, 2021), social psychology (Cikara et al., 2022), and the law (Hu & Kohler-Hausmann, 2024). This perspective aligns with the European approach to combating racism, which largely avoids racial categories in governance, as previously described. However, it is not without challenges. As noted by Braveman and Parker Dominguez, "abandoning the term 'race' has not been accompanied by routine monitoring of health and well-being according to markers of the ethnic groups that are relevant to racism" (Braveman & Parker Dominguez, 2021).[3] This highlights the need for alternative frameworks that capture discrimination while avoiding racial essentialism. What is needed, then, are flexible frameworks with features that can capture discrimination and ensure equal representation based on context-relevant attributes. Following Hu and Kohler-Hausmann, we refer to these as *constitutive features*—attributes that shape the social construct of race—rather than treating race as a fixed variable (Hu & Kohler-Hausmann, 2020). Which features are relevant will depend on the task, the social context, and the geographic setting, leading to a key question: *"Given how a category is constituted, what algorithmic procedures do we consider fair?"* (Hu & Kohler-Hausmann, 2020)

The question above is an interdisciplinary challenge that

needs to be addressed in each deployment context. ML research plays an important role in operationalizing these considerations, providing the tools and frameworks to implement fairness interventions that do not rely on racial taxonomies. Below, we outline a research agenda in ML that does not depend on racial classification, focusing on two types of data: (1) tabular and (2) visual data. We conclude with recommendations on how ML can integrate these approaches, emphasizing participatory AI and interdisciplinary collaboration to ensure fairness frameworks reflect real-world disparities without reinforcing essentialist racial categories.

### 6.1. Tabular Data

Algorithmic fairness research in tabular datasets typically evaluates racial discrimination using predefined, fixed race categories. Some studies assume that these categories are unavailable to auditors ("Fairness under Unawareness") (Chen et al., 2019), yet still treat race as an underlying ground truth inferred from proxies. We argue for a shift in perspective: rather than approximating race through proxies, these indicators should be treated as constitutive features of racial discrimination. This approach shares similarities with causal fairness in ML (Kilbertus et al., 2017), but crucially avoids assuming a causal link between "proxies" and race (Hu & Kohler-Hausmann, 2020), recognizing race as a social construct shaped by context-dependent factors.

Recent work in economics has explored approaches along these lines. For instance, discrimination studies in Swiss online recruitment used a composite ethnicity variable—combining language, nationality, and name-based classification—to analyze hiring biases (Hangartner et al., 2021)[4]. While this method allows for a more direct examination of discrimination, it remains very limited by its reliance on an overlap of three categorical variables, eventually leading again to a binary comparison.

A more flexible alternative has been proposed by Rose (Rose, 2023), replacing categorical race labels with a *race function* —a context-specific mapping of individual characteristics into a racial space, which assigns a percentage of different perceived races to each attribute combination. This approach is in line with our position as it replaces categorical race variables with constitutive features of perceived race. However, the need to define a function that determines such explicit percentages can also be seen as overly specific. Moreover, the approach was proposed for empirical discrimination research rather than auditing ML models. We propose two possible avenues for integrating these insights into algorithmic fairness.

---

[3]It has been found though that perceived race is a better predictor of health disparities than self-reported race (Jones et al., 2008), supporting the focus on racialization.

[4]Name-based classification was based on a name-ethnicity recognition algorithm.

The first avenue consists of connecting this framework to *individual fairness* (Dwork et al., 2012). Individual fairness enforces consistency by ensuring that similar individuals receive similar predictions, based on a task-specific similarity metric. This metric, which operates in the feature space, is conceptually orthogonal to the race function, which instead defines similarity in the space of perceived race. In analogy with independence-based notions of group fairness (Barocas et al., 2023), fairness could be operationalized as the independence of prediction quality or decisions from the position in this space. Exploring connections between the race function approach and individual fairness could, thus, also be seen as a refinement of group fairness along constructivist lines. We stress that the applicability of the individual fairness approach is not straightforward here; indeed, while it overcomes the problem of rigid taxonomies, its problematic reliance on a specific function has been widely discussed in the Fair-ML literature. Instead, we hope that the race function approach, which was more recently developed in Economics (Rose, 2023) and faces very similar problems, can be refined by drawing on this rich literature.

The second route is through *(multi-)calibration* (Hébert-Johnson et al., 2018), which assesses predictive performance across a vast set of attribute-based groups without relying on predefined race categories. This approach comes with the advantage of not relying on specific functions; however, this also means that it does not straightforwardly allow to incorporate the context of domain-specific discrimination or socialization. As a first step, existing multi-calibration algorithms can be adapted to focus specifically on groups defined by relevant constitutive features, ensuring more granular fairness assessments. Future work should better integrate domain knowledge to refine these subgroup definitions and improve how errors are accounted for within these models. Integrating a constructivist understanding of race (Rose, 2023) into a flexible calibration framework (Höltgen & Williamson, 2023) could help address racialization without enforcing discrete groupings.

### 6.2. Visual Data

Computer vision research has increasingly adopted skin tone as a key variable for the study of bias and demographic representation (Buolamwini & Gebru, 2018; Bloomberg, 2023). While this represents progress beyond rigid racial taxonomies, skin tone alone does not fully capture the complexity of racialization and bias in AI (Yucer et al., 2024b).

The idea that racial perception extends beyond skin tone is discussed in the broader literature. Research in cognitive and social psychology shows that racial perception is shaped by multiple phenotypical traits, including skin tone, but also nose shape, eye structure, and lip fullness (Brown Jr et al., 1998; Butler, 2011; Roth, 2016; Travers et al., 2020;

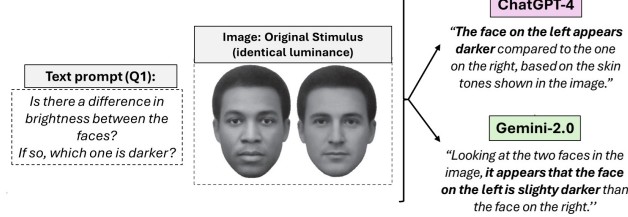

*Figure 3.* Face Race Lightness Illusion (Levin & Banaji, 2006) applied to VQA models (ChatGPT-4o, Gemini-2.0-flash-exp)

Burgund et al., 2024). These traits interact, such that racial categorization is determined by both visual cues and socio-cognitive processes.

A relevant example is the Face Race Lightness illusion (FRL) (Levin & Banaji, 2006), where prototypical faces —what the original study refers to as *Average White* and *Average Black* faces (Levin, 2000)— are perceived differently in terms of lightness, despite having identical luminance. These stimuli were generated by averaging multiple grayscale male faces through an image morphing process, ensuring that the only differences between them lay in their internal features (eyes, nose, and mouth), while overall brightness and contrast remained controlled.

To illustrate that these psychological findings are also relevant in AI, we tested whether Visual Question Answering (VQA) systems exhibit the same perceptual bias (see Appendix E for the details) as humans. We selected ChatGPT-4 (OpenAI, 2023) and Gemini-2.0 (Google AI, 2024), two widely used VQA models, and presented them with the original FRL stimulus (Levin & Banaji, 2006) testing each model ten times (N=10) to assess response consistency. Both models exhibited the perceptual distortion observed in the FRL study (Figure 3). Gemini-2.0 consistently identified one face as darker across all trials, while ChatGPT-4 showed the same pattern in most cases. However, on two occasions (out of 10), ChatGPT-4 did not replicate the illusion and instead reported identical brightness values based on numerical analysis. These results suggest that perceived brightness in these models is influenced by the classification of other facial features, indicating a possible entanglement between phenotypical traits and brightness perception in their latent space. While this example does not imply systematic discrimination, it highlights how facial features can shape skin tone perception, emphasizing the importance of considering such interactions in broader fairness evaluations.

Despite the known influence of phenotypical traits on racial and skin tone perception, the ML community lacks datasets that explicitly annotate diverse phenotypical attributes, and research efforts remain limited and fragmented (Yucer et al., 2024b). For example, the IBM Diversity in Faces

(DiF) dataset (Merler et al., 2019) introduced detailed facial annotations but was later discontinued after it was revealed that the images had been scraped from online sources without explicit consent, raising serious legal and ethical concerns (Harvey, 2021).

The removal of DiF highlights the difficulties of ethically assembling large-scale facial datasets. Given these difficulties, some researchers have re-annotated existing datasets with phenotypical attributes (Yucer et al., 2020; 2022). In particular, (Yucer et al., 2022) proposes a phenotypic-based framework to replace protected attributes like race. This framework considers traits such as skin tone, eyelid type, nose shape, lip shape, hair color, and hair texture—selected based on social behavior (Feliciano, 2016) and medical studies (Fakhro et al., 2015) (see full categories in Appendix F)—allowing them to annotate existing public datasets, such as VGGFace2 (Cao et al., 2018) and RFW (Wang et al., 2019). Their analysis revealed biases in face recognition models, particularly against darker skin tones, wider noses, and monolid eyes, with compounding effects when multiple traits intersect. While this annotation framework has been applied to existing datasets, its use in fairness evaluations remains sparse, and no standardized approach has yet emerged.

To avoid ethical concerns related to collecting phenotypical traits of individuals and to overcome the lack of datasets, some authors have explored the potentials of alternative techniques. GAN-based facial perturbations (Zhang et al., 2022) that manipulate specific features in faces (Yucer et al., 2020; Georgopoulos et al., 2021; Yucer et al., 2024a), and feature-masking (*i.e.*, occluding specific facial regions) (Huang et al., 2023; Ozturk et al., 2024) have been recently explored to isolate and examine the effects of individual phenotypical traits on model decisions.

While recognizing the potential in these works, we highlight that phenotypical attributes are often presented as proxies for race, which can reinforce an essentialist view of race as a biological category (Hanna et al., 2020; Crawford & Paglen, 2021). Instead, they should be considered important components of racial discrimination.

### 6.3. Implementation in practice: Context and Participation

A key challenge in moving away from racial categories is ensuring that fairness interventions remain effective and contextually grounded. A potential solution is the adoption of participatory AI methods, which require direct collaboration with affected communities and domain experts to develop classification criteria and fairness frameworks that reflect specific geographic and socio-political realities. For instance, in Europe, discrimination often targets Romani communities (Trehan & Kóczé, 2009; Fekete, 2014)

and operates through regional hierarchies, such as Mediterranean vs. Nordic identities (Levy, 2015; Macharia, 2017), highlighting the need for fairness frameworks tailored to local contexts. However, meaningful participation requires more than consultation—it must grant real decision-making power to those impacted by AI-driven classification systems (Birhane et al., 2022; Corbett et al., 2023).

Existing research highlights best practices for fostering meaningful community involvement. Birhane et al. emphasize that participatory approaches should center the knowledge and lived experiences of historically marginalized groups, rather than treating them as late-stage consultants in AI development (Birhane et al., 2022). Similarly, Delgado et al. document a participatory AI case study in the legal field, where interdisciplinary collaboration between domain experts and computer scientists led to iterative system design improvements, enhancing both effectiveness and fairness (Delgado et al., 2022). However, these efforts must be carefully implemented to avoid "participation-washing"—where community involvement remains superficial and fails to shift power dynamics—can undermine the intended impact of these initiatives (Sloane et al., 2022).

We propose the following 4 principles, corresponding to the acronym EDTAL, to ensure that participatory AI leads to meaningful fairness interventions: (1) Early and sustained **E**ngagement – community participation should begin at the problem-definition stage, not just during model evaluation; (2) **D**ecision-making power – affected groups should play an active role in shaping AI development, beyond mere consultation; (3) **T**ransparency and **A**ccountability – the rationale and methods for community involvement should be well-documented and publicly accessible to prevent tokenism; and (4) **L**ocalized and context-sensitive approaches – considering regional and cultural differences, as racial biases manifest differently across contexts (Ball et al., 2022; Fekete, 2014).

## 7. Alternative Views

The most obvious critique of our proposal is the position that race categories are too important and/or useful to eliminate them. A first argument may especially be raised by those who identify with a marginalized racial group and worry that omitting the categories limits their visibility and their ability to have a voice. A related reasoning was expressed in the hearings following the U.S. decision to allow the indication of multiple races, with groups within the civil rights movement arguing against this plan, based on worries that it may reduce the political effectiveness, despite rejecting the existence of separate biological races (Robbin, 2000; Williams, 2003). Such a position is typically referred to as *strategic essentialism*, a "political strategy whereby differences (within a group) are temporarily downplayed and unity assumed for

the sake of achieving political goals" (Eide, 2016). Since the 60s, U.S. law now also explicitly provides protection against disparate impact across racial groups in specific contexts, most notably employment (Barocas & Selbst, 2016). Similar arguments to strategic essentialism may be made for the necessity of race categories in ML, for pointing out discrimination in applications like COMPAS.

We think that strategic essentialism is indeed a legitimate and sometimes necessary strategy, as different constituents of race are adequate in different contexts (Roth, 2016). However, the important nuance we would like to stress in this paper is that racial categories in the context of ML should neither be imposed from outsiders in a position of power nor be considered universally applicable. Furthermore, there should be a critical discussion about their construction and a justification for their use (Mitchell et al., 2019; Okoro et al., 2021; Gebru et al., 2021; Crisan et al., 2022; Mickel, 2024). Discrimination studies —such as the analysis of the COMPAS system— often simply interpret race as "perceived race" which does not do justice to complexity of the concept (Hu & Kohler-Hausmann, 2024). Regarding the bearing of disparate impact laws in the U.S. on ML algorithms, there is a concern from the legal perspective "that developers will focus too narrowly on [Statistical Parity Tests], making choices keyed to these metrics, rather than try to understand why disparities are arising and where substantive unfairness may be affecting the selection process" (Raghavan & Kim, 2024). While demographic parity between race categories can already be subject to substantial distribution shifts across different states within the United States (Ding et al., 2021), such failures for fairness criteria to generalize can be expected to be even stronger across geographic regions with substantially different historical biases experienced by racial groups. In sum, our position is that ML researchers and practitioners should be more critical of simplistic race taxonomies and develop methods to analyze discrimination more flexibly, even if there can be situations where their use is warranted.

## 8. Conclusion

In this paper, we advocate for the abandonment of race categories in machine learning by default, and call for a fundamental rethinking of how race is conceptualized and operationalized. Building on research from other disciplines, we discuss the ethical, societal, and philosophical challenges of treating race as a categorical "ground truth" both in tabular and visual data. We contrast U.S.-centric existing race taxonomies with practices in Europe and illustrate in the case of mixed-race individuals how current AI practices fail to address the complexities of racial identity while risking to harm marginalized communities. Further highlighting the problems of reifying and stereotyping race in AI, we call

for alternative approaches and suggest research directions for ML. Detecting discrimination is not merely a formal computational task, but also a contextual and normative endeavor (Selbst et al., 2019; Wachter et al., 2021; Hu & Kohler-Hausmann, 2024). Thus, moving beyond fixed racial categories requires careful scrutiny to ensure that fairness interventions remain effective without reinforcing essentialist assumptions. We call on the ML community to critically engage with these challenges and develop the awareness and tools necessary for a more nuanced and equitable approach.

## Acknowledgements

MD acknowledges support from the ARIAC project (No. 2010235), funded by the Service Public de Wallonie (SPW Recherche), and funding from the FNRS (National Fund for Scientific Research) for her visiting research at the ELLIS Alicante Foundation.

BH is supported by the German Federal Ministry of Education and Research (BMBF): Tübingen AI Center, FKZ: 01IS18039A. BH also acknowledges support by the International Max Planck Research School for Intelligent Systems (IMPRS-IS) and travel support from ELIAS (GA no 101120237).

PR and NO are supported by a nominal grant received at the ELLIS Unit Alicante Foundation from the Regional Government of Valencia in Spain (Convenio Singular signed with Generalitat Valenciana, Conselleria de Innovación, Industria, Comercio y Turismo, Dirección General de Innovación). PR is also supported by a grant by Fundación Banc Sabadell.

We thank all those who shared their time and thoughts with us during the early stages of this work, especially members of the mixed-race community, whose perspectives deepened our understanding of racial categorization and the nuanced experiences of in-betweenness.

## Impact Statement

This paper argues that the use of race categories in machine learning research and applications should be abandoned. Indeed, we propose the adoption of alternative features relevant for diversity that are better suited to account for complex, multidimensional aspects of human identity and social inequalities.

We acknowledge that this approach may have significant societal implications. While we seek to advance equity and fairness in AI, our approach also requires careful consideration of the features used to ensure they accurately and responsibly represent diversity without introducing new biases or unintended consequences. To this end, we advocate for interdisciplinary collaboration, transparency, and rigorous evaluation of diversity metrics to ensure ethical and

effective implementation.

**Author's positionality**

We recognize the importance of reflecting on our own identities and experiences in shaping this work. Our team consists of four authors—three identifying as female and one as male—with three different European nationalities. Collectively, we have lived, studied, and worked across various international contexts, both inside and outside Europe, which informs our understanding of global perspectives on race and technology. The first author, being of European-African heritage, brings personal insight into the lived experiences of navigating racial identity, directly informing the critical analysis presented in this paper. Our combined expertise spans over 30 years of research in artificial intelligence, computer vision, human-computer interaction, AI for social good, algorithmic fairness, and philosophy of science, grounding our exploration of race taxonomies in AI within both technical and ethical frameworks. We acknowledge the limitations of our perspectives and remain committed to ongoing reflection and engagement with the communities most impacted by these technologies.

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

## A. Race-Related Datasets used in ICML and CVPR Papers (2023-2024)

Table of the 28 accepted papers that incorporate race categories in their research from ICML (featuring 'fair' in the title) and CVPR (featuring 'fair' or 'bias' in the title), along with the datasets utilized.

*Table 1.* List of race-related datasets used in fairness research, indicating the papers and their respective conferences (CVPR = * and ICML = *). (2023-2024)

| Race Dataset | Papers Using the Dataset *= ICML, *=CVPR (2023-2024) |
| --- | --- |
| FairFace (Karkkainen & Joo, 2021) | [(Chen et al., 2024), (Qraitem et al., 2023), (Garcia et al., 2023), (D'Incà et al., 2024)]* |
| UTKFace (Zhang et al., 2017) | [(Qraitem et al., 2023), (Park & Byun, 2024)]*; [(Nelaturu et al., 2024)]*: |
| COMPAS (ProPublica, 2016) | [(Soen et al., 2023),  (Memarrast et al., 2023), (Singh et al., 2023), (Zhu et al., 2023), (Peltonen et al., 2023), (Roh et al., 2023), (Kim & Zubizarreta, 2023), (Becker et al., 2024), (Tifrea et al., 2024)]* |
| Census (Adult / Folktables) (Ding et al., 2021) | [(Xian et al., 2023), (Khalili et al., 2023), (Jovanović et al., 2023), (Singh et al., 2023),(Okati et al., 2023),(Knittel et al., 2023),(Keswani et al., 2024)]* |
| MINNEAPOLIS [5] | [(Soen et al., 2023)] * |
| FASSEG (Khan et al., 2015) | [(Zhang et al., 2024)]* |
| HSLS (Jeong et al., 2022) | [(Tifrea et al., 2024)]* |
| ENEM (Alghamdi et al., 2022) | [(Tifrea et al., 2024)]* |
| Law School (Wightman, 1998) | [(Peltonen et al., 2023),(Xu & Strohmer, 2023),(Xian et al., 2024)] * |
| Communities & Crime (Wightman, 1998) | [(Xian et al., 2023),(Singh et al., 2023),(Xu & Strohmer, 2023),(Giuliani et al., 2023),(Xian et al., 2024), (Tifrea et al., 2024)] * |
| Toxic Comments (Borkan et al., 2019) | [(Xu et al., 2024)] * |

## B. U.S. census categories

Since 1997, race/ethnicity data was collected in the U.S. census as depicted in Figure 4 (US Office of Management and Budget, 1997). In 2024, the Statistical Policy Directive No. 15 on Race and Ethnicity Data Standards was amended for the first time since 1997. Among other changes, a new category "Middle Eastern or North African" was added, which was previously subsumed under "White".

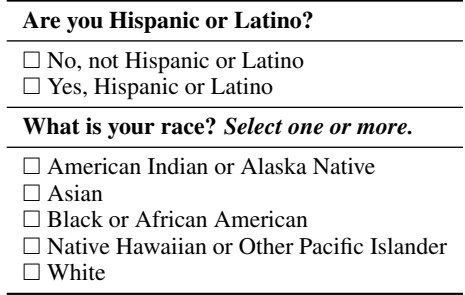

*Figure 4.* U.S. census questions for race and ethnicity after 1997.

## C. Additional information on datasets

Table 2. Race categories utilized in popular Machine Learning datasets. More information in Appendix C.

| Dataset | #Categories | U.S. Census subset | Note |
|---|---|---|---|
| COLORFERET | 4 | √ | |
| MORPH | 4 | √ | |
| UTK FACE | 5 | √ | *Indian* separated from *Asian* |
| FAIRFACE | 7 | √ | *White* & *Asian* subdivided, *Indian* separated |
| ADULT / FOLKTABLES | 5/9 | √ | |
| COMPAS | 6 | √ | |
| COMMUNITIES & CRIME | 4 | √ | |
| LSAC LAW SCHOOL | 8 | × | plus *Mexican American* & *Puerto Rican* |

ColorFERET (Phillips et al., 2000) use *White, Asian, Black, Others*

MORPH (Ricanek & Tesafaye, 2006) use *Caucasian, Hispanic, Asian, African American*

UTKFace (Zhang et al., 2017) use *Asian, Black, Indian, White, Others*

FairFace (Karkkainen & Joo, 2021) use *Black, East Asian, Indian, Latino, Middle Eastern, Southeast Asian, and Western*; the authors follow the U.S. census standard to subdivide White into Western and Middle-Eastern, but see Asian as subdivided into East Asian and Southeast Asian, with Indian as an independent race, contrary to the U.S. census.

Adult / folktables (Becker & Kohavi, 1996; Ding et al., 2021) is directly derived from census data (with mixed-race coded as single category); the old Adult data only had *White, Asian-Pacific-Islander, American-Indian-Eskimo, Other, Black*. In the new folktables version, the full category names are *White, Black or African American, American Indian, Alaska Native, American Indian or Alaska Native*[6], *Asian, Native Hawaiian and Other Pacific Islander, Other Race, Two or More Races*

COMPAS (ProPublica, 2016) uses *African-American, Asian, Caucasian, Hispanic, Native American, Other* based on police/judicial data

Communities & Crime (Redmond & Baveja, 2002) uses U.S. census data for the percentages of race categories per community, using only *black, white, asian, hispanic*

LSAC Law School (Wightman, 1998) divides by "Ethnic Group" categories and uses *American Indian, Asian American, Black, Mexican American, Puerto Rican, Hispanic, White, Other*

For all the tabular datasets, it is also common to binarize into white/non-white or black/non-black (in the case of Communities & Crime) or to use any other subset. This is despite the Statistical Policy Directive No. 15 from 1997 declaring that "[t]he term 'nonwhite' is not acceptable for use in the presentation of Federal Government data."

## D. Mixed-Race Identity

**Definition of "mulatto"** The inclusion of the term *"mulatto"* in this study was intentional and motivated by its definition in the Oxford English Dictionary (Oxford English Dictionary, 2025a).

> "mulatto, n. & adj. A person with one white and one black parent. Frequently more generally: a person of mixed white and black ancestry. Cf. metis, n. A.1, quadroon, n. Now chiefly considered offensive."

This term, while offensive in contemporary English, reflects a historical framing of mixed-race identities within a narrow Black/White binary. We hypothesize that Generative AI systems, trained on datasets infused with historical and cultural biases, may replicate or reinforce such reductive categorizations.

---

[6]Full: American Indian and Alaska Native tribes specified, or American Indian or Alaska Native, not specified and no other races

**Additional information regarding the Midjourney and Stable-Diffusion 3.5 Large experiments**    To investigate the representation of mixed-race individuals, we conducted an experiment using Stability AI's Stable Diffusion 3.5 Large (Huggingface, 2024), Midjourney (MidJourney, 2024), and OpenAI's DALL·E 3 (OpenAI, 2024). The experiment focused on two specific prompts: *"a mixed-race person"* (MR-prompt) and *"a mulatto person"* (Mu-prompt). For each prompt, we generated 30 images per model, resulting in two distinct datasets for comparison.

As real images, we used the 88 images categorized as "Mixed-Race" in the Chicago Real Face Database (Ma et al., 2021). To ensure consistency, all real and generated images were cropped before feature extraction, focusing on the facial area

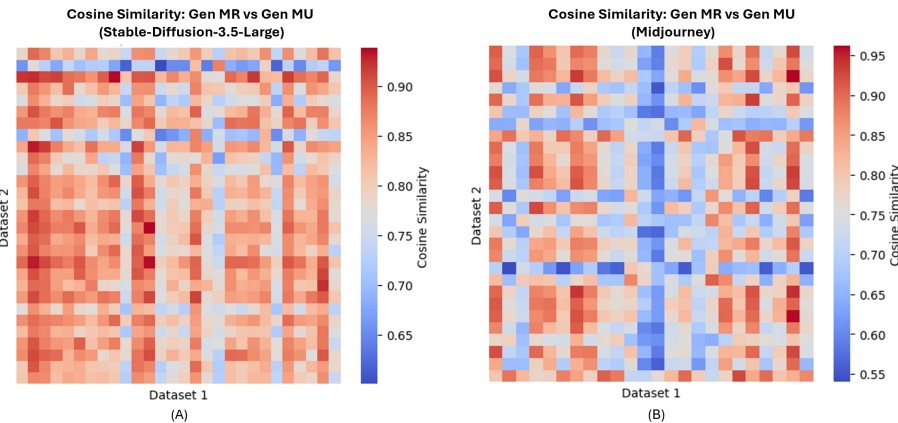

*Figure 5.* Heatmap of cosine similarity between images generated using the prompts *"mixed-race person"* (MR-PROMPT) and *"mulatto person"* (MU-PROMPT). On the left (A), results from Stable Diffusion 3.5-Large; on the right (B), results from MidJourney.

*Table 3.* Cosine similarity distribution between images generated for "a mixed-race person" (MR-PROMPT) and " a mulatto person" (MU-PROMPT) in Stable Diffusion and MidJourney.

| Model | Mean | Median | 75th Percentile | 90th Percentile |
|---|---|---|---|---|
| Stable Diffusion | 0.8190 | 0.8253 | 0.8618 | 0.8849 |
| MidJourney | 0.7687 | 0.7732 | 0.8228 | 0.8975 |

*Table 4.* Cosine similarity statistics for real and AI-generated mixed-race images. The generated images (SD = Stable Diffusion-3.5-Large and MDJ = MidJourney) exhibit higher internal similarity compared to real-world images from CFD-MR, indicating a lower degree of phenotypic diversity.

| Dataset | Mean | Median | 75th Percentile | 90th Percentile |
|---|---|---|---|---|
| Real Mixed-Race (CFD-MR) | 0.7883 | 0.7911 | 0.8489 | 0.8831 |
| Generated Mixed-Race (SD) | 0.8694 | 0.8770 | 0.9050 | 0.9272 |
| Generated Mixed-Race (MDJ) | 0.8265 | 0.8388 | 0.8933 | 0.9209 |

## E. Additional information on testing the Face Race Lightness Illusion in VQA Models

**Original test:**    (Levin & Banaji, 2006) tested this effect by using grayscale images of faces, carefully controlled for luminance, and asked human participants to adjust the brightness of one face to match another. Their results showed that participants consistently perceived Black faces as darker than White faces, even when their objective brightness was the same. This suggests that phenotypical traits can override purely physical visual cues in human perception.

**Adaptation:**    Inspired by these findings, we investigated whether Visual Question Answering (VQA) models exhibit similar perceptual distortions when processing faces with different phenotypical traits but identical luminance. While AI

models cannot adjust brightness levels as human participants did in the original study, they can evaluate and compare brightness differences based on learned visual representations. Thus, our goal was to examine whether VQA models display a bias in perceived brightness similar to that observed in humans. (models: OpenAI's ChatGPT-4 (OpenAI, 2023), Google's Gemini-2.0-flash-exp (Google AI, 2024)). We employed the same stimulus used in the original FRL experiment (Levin & Banaji, 2006). Specifically, we presented the models with the validated pair of average men faces from (Levin & Banaji, 2006) 3.

Our choice to use this specific stimulus ensures a direct replication of the original study in an AI context, avoiding the need for additional psychological validation that new stimuli would require.

We asked each model the following question: **Q1:***"Is there a difference in brightness between the faces? If so, which one is darker?"* . This query was repeated ten times **(N=10)** per model to assess response consistency.

- Gemini-2.0 consistently identified the left face as darker in all ten trials, fully replicating the perceptual distortion observed in human participants.

- ChatGPT-4 showed a similar pattern in eight out of ten trials but deviated twice, providing an analytical response: *"The average brightness of the left face is approximately 196.32, while the average brightness of the right face is about 196.32. Both faces appear to have identical brightness levels."* This suggests that ChatGPT-4 sometimes relied on numerical analysis, which overrode the perceptual distortion.

To further investigate the reasoning behind the models' responses, we asked: **Q2:***"Why do you think the left face appears darker?"* . Both models attributed the effect to differences in luminance and contrast, but with slight differences in emphasis. Gemini 2.0 provided a more structured explanation, explicitly citing melanin concentration and grayscale rendering as primary factors, while ChatGPT-4 offered a more general response, framing the effect in terms of brightness differences and perceptual biases.

"The left face appears darker primarily due to the amount of pigment in the skin. This is usually due to a higher concentration of melanin, which is a natural pigment responsible for skin, hair, and eye color. In the case of these images, the left face likely belongs to someone with darker skin, leading to a darker appearance in the grayscale rendering of the image.

Here are some supporting points:

- Melanin Content: The amount of melanin in an individual's skin is the primary factor determining skin darkness. People with darker skin tones have higher levels of melanin.
- Grayscale Imaging: When color images are converted to grayscale, the brightness level of each pixel corresponds to the intensity of the color. Higher levels of pigment (like melanin) absorb more light, resulting in a darker pixel in the grayscale image.
- Visual Perception: Our eyes and brains interpret darker areas as having less light reflecting off them, and consequently, we perceive them as darker.

Therefore, the darker appearance of the left face is primarily due to the higher concentration of melanin in the individual's skin, which translates to a darker shade in the grayscale image." (Example:*Gemini 2.0*)

"The left face appears darker due to differences in luminance and contrast. The grayscale shading used in the image likely has lower brightness levels on the left face compared to the right. Additionally, our perception might be influenced by cognitive biases and expectations related to facial features and shading." (Example:*ChatGPT-4*)

## F. Phenotypic-based framework

Phenotypical attributes used by (Yucer et al., 2022) to label VGGFace2 and RFW datasets inspired by the relevant categories in (Feliciano, 2016).

*Table 5.* List of phenotypical attributes and their respective categories used in (Yucer et al., 2022)

| Attribute | Categories |
| --- | --- |
| Skin Type | Type 1 / 2 / 3 / 4 / 5 / 6 (Fitzpatrick Skin Types (Sachdeva, 2009)) |
| Eyelid Type | Monolid / Other |
| Nose Shape | Wide / Narrow |
| Lip Shape | Full / Small |
| Hair Type | Straight / Wavy / Curly / Bald |
| Hair Colour | Red / Blonde / Brown / Black / Grey |

