# OpenReview forum: "Position: The Categorization of Race in ML is a Flawed Premise"
_ICML.cc/2025/Position_Paper_Track — ICML 2025 Position Paper Track spotlightposter_

### Official Review · Reviewer_1qrB · 2025-03-09

**Significance:** 4
**Argument Clarity:** 4
**Ethics Flag:** Yes
**Rating:** 5
**Confidence:** 5

**Questions:**

No questions.

**Discussion Potential:**

4

**Paper Summary:**

In this paper, the authors question the reliance of ML fairness literature on a discrete set of race categories. The authors state that these categories are based on social, ill-practiced constructs and are not biologically justified. Moreover, these categories do not capture the varieties of races encountered in the society.

## update after rebuttal

I've read the comments by other reviewers and the rebuttal provided the authors. I recommend the acceptance of the paper.

**Position:**

Yes

**Position In Title:**

Yes

**Related Work:**

4

**Strengths And Weaknesses:**

Strengths:
+ The paper highlights a very fundamental problem in ML fairness.
+ It provides a very good summary and background for the problem and its effects.
+ It proposes a reasonable roadmap to address the issue.

Weaknesses:

I am very happy with the paper. As a senior researcher working on bias & fairness for a while, I was questioning the praticality and the effectiveness of ML fairness based on the race categories provided with the datasets.

**Support:**

4

---

> ### Author Rebuttal · Authors · 2025-03-29
>
> We sincerely appreciate your thoughtful and encouraging feedback. It is especially meaningful and encouraging to read that our research resonates with someone who is deeply familiar with these issues after years of research in this field. Thank you very much for your time and insights.

---

### Official Review · Reviewer_TbEj · 2025-03-11

**Significance:** 3
**Argument Clarity:** 3
**Rating:** 4
**Confidence:** 3

**Questions:**

1. How can we avoid unintentionally overlooking or neglecting differences in these areas while ensuring fairness when moving away from racial classifications?
2. Regarding the alternative methods proposed for tabular data, such as connecting the race function approach with individual fairness, could you further discuss the challenges in practical implementation and how to overcome these challenges?

**Discussion Potential:**

3

**Paper Summary:**

This paper critiques the use of rigid racial classifications in machine learning (ML), arguing that these classifications fail to capture the complexity of race. It proposes avoiding categorical race labels by integrating context and domain knowledge, in order to more effectively address discrimination issues in the ML field. Specifically, the paper's contributions include the following aspects:
1. The paper analyzes the problems caused by the use of racial classifications (such as "White," "Black," "Asian," etc.) in current machine learning, particularly how these classifications are often limited to U.S.-centric social and cultural frameworks. These labels fail to effectively reflect the complexity of racial identities, overlook the diversity of mixed-race populations, and reinforce outdated essentialist views.
2. The paper approaches race from a sociological and historical perspective, arguing that race is not a biologically discrete category but rather a dynamic concept shaped by social environment, cultural background and personal experience.Therefore, existing ML racial classification methods cannot effectively capture the complexity of race。
3. The paper proposes a research agenda that moves beyond racial labels in machine learning.  It advocates treating discrimination-related features as "constitutive features," and explores how to use context and domain knowledge in machine learning to more effectively solve problems of discrimination and social inequality.

**Position:**

Yes

**Position In Title:**

Yes

**Related Work:**

3

**Strengths And Weaknesses:**

Advantages:
1. The paper’s position is supported by substantial evidence from multiple disciplines, including anthropology, sociology, and computer science.
2. The topic is highly relevant to the ICML community, as it addresses fairness, bias, and the handling of sensitive data in machine learning.
3. It has the potential to spark discussion, as it challenges the common practice of using racial categories in machine learning. It also discusses alternative perspectives, such as arguments related to strategic essentialism.
4. The paper’s argument is clear, referencing a wide range of relevant works. It presents a logical flow, from critique of current practices to proposed solutions.

Disadvantages:
1. The proposed alternatives, such as treating discrimination-related features as constitutive rather than categorical, may be difficult to implement in real-world ML systems.
2. Although the argument is strong, there is a lack of real-world examples directly testing the proposed framework.

**Support:**

3

---

> ### Author Rebuttal · Authors · 2025-03-29
>
> We thank the reviewer for the insightful comments and the positive evaluation of our paper. Below, we answer the questions and comment on the weaknesses raised in the review.
>
> Q1. Thank you for the comment. While it is true that implementing constitutive features in real-world ML systems presents challenges, these difficulties are not insurmountable. In this paper, we argue that racial discrimination is not driven by an inherent, independently existing notion of race but rather by racialisation, a process that can be captured through specific context-dependent cues (“constitutive features”). For example, in the cited work of (Hangartner, 2021), hiring discrimination is examined based on factors such as language, nationality, and ethnic origin of the name rather than using a race category. While this approach is relatively simplistic --comparing only a fully Swiss group with a fully non-Swiss group-- it highlights how racialization operates through specific attributes rather than an abstract race label.
>
> Building on insights from the algorithmic fairness research, ML systems can better capture these complexities and the interplay between the features, including intersectionality, by employing techniques such as multicalibration across all relevant subgroups formed by interactions of constitutive features. Existing multicalibration algorithms (Hebert-Johnson et al., 2018) can be adapted to focus specifically on groups defined by these features, ensuring more granular fairness assessments. Future work should better integrate domain knowledge to refine these subgroup definitions and improve how errors are accounted for within these models.
>
> As noted in the paper, we do not claim that race categories are not useful in some circumstances and use cases. However, we emphasize the importance of situating their use within a well-justified and context-specific framework.
>
> Q2. One of the main challenges in implementing our proposed approach is the availability of relevant constitutive features and the need to account for their complex interdependencies. However, these challenges are not necessarily greater than those associated with using race categories in ML models. In many cases, race data is unavailable, especially – but not only – in Europe, but fairness and discrimination auditing methods have still been developed in such contexts.
>
> To address the challenge of feature availability, existing research on fairness under unawareness offers useful strategies for identifying and leveraging proxy variables when direct group membership information is missing. The methods can be adapted to work with constitutive features, ensuring that relevant information about racialization is included without relying on race categories. The second challenge -capturing the complex interplay between constitutive features- can be addressed using flexible approaches from the ML literature, such as multicalibration, which ensures that predictions remain fair across all meaningful subgroups defined by interactions of constitutive features, rather than assuming a single and fixed group structure.
>
> While these approaches provide a starting point, further research is needed to refine existing methods and integrate domain knowledge about how constitutive features interact in specific social contexts.
>
> **Weakness**:
>
> W1. In the case of Computer Vision, there is previous work showcasing the feasibility of the proposed approach, as outlined in the paper. We discuss the case of tabular data in our responses to Q1 and Q2. An important contribution of this position paper is a call for more research on these directions.
>
> W2.  Real-world validation is essential to assess how the proposed ideas perform in real-world applications, especially in high-stakes areas like hiring, healthcare, or criminal justice. While there is a lack of extensive case studies, this should not be seen as a fundamental flaw but rather as an opportunity for future research and experimentation.
> The adoption of these concepts in real-world scenarios is an evolving process, and as fairness-aware machine learning becomes more mainstream and awareness grows, we anticipate more real-world examples that align with our work to emerge.
>
> Notably, recent initiatives in fairness-aware healthcare modeling have started to integrate structural factors, such as access to care and economic background, instead of directly relying on race. These efforts provide early evidence of how constitutive features can be operationalized in practice. Similarly, research in hiring discrimination has explored using nationality, language and name origin as proxies for racialization rather than using broad racial categories, demonstrating the feasibility of the proposed framework (Hangartner, 2021).
>
> We acknowledge that further empirical validation will be necessary and we hope that our work will encourage more researchers and practitioners to test and refine the proposed approach in real-world settings.

---

> > ### Comment · Reviewer_TbEj · 2025-04-08
> >
> > Thank the authors for the response. I maintain the score as 4-accept.

---

### Official Review · Reviewer_6ati · 2025-03-14

**Significance:** 3
**Argument Clarity:** 3
**Rating:** 4
**Confidence:** 4

**Questions:**

Please address the weaknesses above.

**Discussion Potential:**

2

**Paper Summary:**

The authors argue that the use and categorization of race in ML is flawed. Their arguments are that (1) racial taxonomies are US-centric, and race is not a legitimate classification category in Europe, (2) existing racial categorizations do not adequately account for mixed-race individuals, and (3) codifying race may reinforce stereotypes in ML systems. As alternatives, the authors propose to use other fairness definitions, to incorporate context domain knowledge, and to engage the community.

**Position:**

Yes

**Position In Title:**

Yes

**Related Work:**

3

**Strengths And Weaknesses:**

Strengths:
- The topic is important and under-explored.
- The mixed-race argument is compelling.
- The authors support their argument with prototype experiments on foundation models.


Weaknesses:
1. The authors should address one important counterargument from the ML modeling perspective, which is that using race as an input feature not only improves overall model performance, but also the performance of each racial group, especially in healthcare [e.g. 1-2]. While I completely agree that race is being used as a proxy here for unobserved variables (e.g. socioeconomic status), which may cause brittleness under distribution shift, the use of race in this case does give a tangible benefit when crucial causal variables are not known.

2. I think that the argument of distribution shift/transferrability is a strong argument for excluding race in ML models, which the authors should add to their work. In particular, the historical biases faced by marginalized groups in the US are quite different than the biases faced by the same racial groups in other countries. Due to this context dependence, models performance across racial groups, or models trained to equalize such performance, may not transfer meaningfully across different regions or time periods.

3. In my opinion, one reason why race is so common in benchmark data from the US is that race is encoded as a protected group in US federal law. Presumably this has some implications for the degree to which deployed models can vary their predictions by race, and so model developers need to evaluate performance across race, and try to mitigate biases across race whenever possible. The authors should discuss the impact of race as a legal protected group (e.g. the 4/5th rule) on ML model development and fairness evaluations.

4. I'm not convinced by the alternatives proposed by the authors. For example, multicalibration and individual fairness are proposed as alternatives in the case of tabular data, but they are very different definitions from conventional group fairness, and have their own drawbacks. Similarly, the authors argue for community engagement and participatory AI, which is admirable, but is infeasible in many cases and would increase cost of data collection.

5. One counter-argument is that in ML fairness benchmark datasets such as ACS, the use of race is purely academic, i.e. it is a categorical variable where we see disparate performance across subsets of this variable, and could be exchanged for any other categorical variable where this occurs. Papers in ACM FAccT which build or evaluate models using race are using it simply to test the efficacy of their algorithms (along with other arbitrary categorical variables such as age and sex), and these models are not being deployed in the real world, and so do not present any real harm. How would the authors respond to this?


[1] Race adjustments in clinical algorithms can help correct for racial disparities in data quality. PNAS 2024.

[2] Use of race in clinical algorithms. Science Advances 2023.

**Support:**

2

---

> ### Author Rebuttal · Authors · 2025-03-29
>
> We thank the reviewer for the insightful suggestions and questions, the constructive feedback and the positive evaluation of our paper. Below, we respond to the 5 weaknesses identified in the review.
>
> **W2**: This is a very good point. We will include this argument in the revised version of the paper. Indeed, the use of race is not robust to distribution shifts. As you very well note, the historical biases faced by marginalized groups in one country (e.g. the US) are different from the biases faced by the same racial groups in other countries. This problem is easily overlooked in a simplistic treatment of race categories but it fits well and can be incorporated into the paper to further strengthen our position.
>
> **W1**: Race can improve model performance in healthcare, but it is a shortcut rather than a solution. Using race as a feature raises important ethical and legal concerns, it is fragile under distribution shifts (as previously noted), it does not address the root causes for disparities and it lacks precision as a predictive feature in general terms. See also footnote 3 in the paper.
>
> **W3**: We agree that the legal status of race as a protected category in the US has shaped fairness evaluations. US law provides protection against disparate impact across racial groups,  but also requires considering context and there is a concern from the legal perspective  *“that developers will focus too narrowly on [Statistical Parity Tests], making choices keyed to these metrics, rather than try to understand why disparities are arising and where substantive unfairness may be affecting the selection process”* [1].  As stated in the paper, the EU legal landscape --including GDPR and national anti-discrimination laws-- tends to limit the collection and use of racial data. We argue that fairness ML research should develop methods for mitigating discriminatory patterns without relying on racial labels.
>
> **W4**: Indeed existing methods on tabular data are not yet satisfying and our position paper calls for more research. Regarding multicalibration, please see our response to Q1 by Reviewer TbEj. The applicability of individual fairness (IF) is more unclear: the idea is to take the existing literature as an inspiration to refine the race function approach (Rose, 2023) suggested in Economics. This approach promises to overcome the problems of rigid taxonomies but relies on a specific function — a drawback shared by IF, such that insights from work in ML on this, spanning over a decade, can be leveraged. While community engagement and participatory AI may introduce practical challenges and increased costs, we view them as long-term investments that yield more contextually aware and socially grounded ML systems rather than short-term fixes. Studies such as Birhane et al. (2022) and Delgado et al. (2022) demonstrate that even partial participatory approaches can significantly enhance fairness, accountability, and contextual relevance. We welcome further discussion on how to scale these methods across diverse contexts, as these investments are essential for developing more equitable AI practices.
>
> **W5**: While many ML fairness benchmarks treat race as a categorical variable—much like age or sex—for testing algorithmic performance, this approach overlooks several key points: 1. Unlike age—a continuous, objective measure—racial categories carry deep historical, socio-cultural, and political significance, reflecting legacies of colonization and discrimination. Treating race as interchangeable risks oversimplifying complex societal issues.
> 2.  Although many papers using race in benchmarks are for testing algorithmic efficacy, the methodologies, fairness metrics, and insights developed in academic settings often shape the design and evaluation of systems that eventually impact society. If the underlying assumptions about race are flawed or uncritically adopted, this can perpetuate harmful biases when these models are later applied in practice. 3. Even if using race in benchmarks is primarily academic, it contributes to the broader discourse on fairness and representation. Misrepresenting or oversimplifying race can lead to an erasure of the lived experiences of communities affected by historical and ongoing discrimination. As discussions at ACM FAccT have highlighted, there is a strong call within the community to contextualize and document the use of racial categories rather than treating them as neutral.
> Our position doesn’t preclude using racial categories when necessary, but calls for their use with careful contextualization and justification—a level of reflection that is often missing in current practice.
>
> Again, we thank the reviewer for the insightful comments and constructive feedback. We will adapt the revised version of the paper accordingly.
>
> [1] Raghavan, Manish, and Pauline T. Kim. "Limitations of the" Four-Fifths Rule" and Statistical Parity Tests for Measuring Fairness." Geo. L. Tech. Rev. 8 (2024): 93.

---

> > ### Comment · Reviewer_6ati · 2025-04-02
> >
> > Thank you for the response. Most of my concerns have been addressed, and I will raise my score to a 4.

---

> > > ### Author Response · Authors · 2025-04-04
> > >
> > > Thank you very much!!

---

### Official Review · Reviewer_uy1p · 2025-03-15

**Significance:** 3
**Argument Clarity:** 3
**Rating:** 3
**Confidence:** 4

**Questions:**

1. How would you operationalize “constitutive features” in practice? Could you provide a concrete example for a specific application (e.g., hiring algorithms)?
2. The paper critiques the EU’s avoidance of racial categories but does not address how your proposed methods align with GDPR restrictions. How can ML systems balance anti-discrimination goals with legal constraints on racial data collection?
3. The generative AI experiment focuses on “mixed-race” and “mulatto” prompts. Why were other mixed-race identities (e.g., Asian-White) not included, and how might this affect the conclusions?

**Discussion Potential:**

3

**Paper Summary:**

This paper critiques the use of categorical race labels in machine learning, arguing they oversimplify racial identity, ignore mixed-race experiences, and perpetuate stereotypes. It advocates for more context-specific, interdisciplinary approaches to address discrimination, emphasizing participatory AI and domain knowledge.

**Position:**

Yes

**Position In Title:**

Yes

**Related Work:**

3

**Strengths And Weaknesses:**

Strengths:
The paper presents a timely and clear argument on the ethical implications of racial labels in ML, supported by interdisciplinary insights and empirical evidence, while offering actionable alternatives like constitutive features and context-sensitive frameworks to move beyond categorical racial classifications.

Weaknesses:
1. Limited technical detail: The paper lacks specifics on how to operationalize constitutive features or integrate them into existing fairness frameworks (e.g., metrics, algorithms).
2. Narrow focus: The contrast between U.S. and European approaches is emphasized, but broader global perspectives (e.g., Asia, Africa) are underexplored, limiting the universality of the argument.
3. Robustness issues: The CFD-MR and generative AI experiments use small sample sizes (e.g., 88 mixed-race individuals), raising questions about statistical robustness.

**Support:**

3

---

> ### Author Rebuttal · Authors · 2025-03-29
>
> We thank the reviewer for the insightful comments and the positive evaluation of our paper. Below, we provide an answer to the 3 questions posed in the review.
>
> Q1: Constitutive features should reflect the underlying social, economic, and historical factors that can contribute to discrimination and disparities in outcomes in a given context, without relying on socially constructed racial categories. For computer vision, there are existing approaches while for tabular data, we point to research directions that we consider promising. We elaborate on the hiring case in our response to Q1 by Reviewer TbEj.
>
> Q2: We would like to clarify that our paper does not critique the EU’s avoidance of racial categories. Our proposal is aligned with regulation and legislation that restricts the use of racial data as we propose that such data should not be used to train and evaluate ML algorithms. According to the Court of Justice of the European Union, ‘ethnic origin cannot be determined on the basis of a single criterion’ and judicial decisions on racial or ethnic discrimination need to consider multiple factors [1]. Different member states have different standards on which attributes (constitutive features) are relevant for racial discrimination [2], highlighting the importance of contextuality. While for some applications, some attributes may be considered sensitive under the GDPR, our proposal is more aligned with it than the conventional approach that relies on race labels.
>
> Q3: Our decision to focus on the prompts 'mixed-race' and 'mulatto' is intentional and grounded in their well-documented historical context which provides a clear reference for identifying stereotypical representations. The term 'mulatto,' while controversial today, carries a specific legacy that allows for a more grounded critique of how generative models replicate racial stereotypes. By contrast, prompts like 'Asian-White' lack a similarly established cultural reference, making it difficult to assess whether the outputs reflect stereotyping or natural variation. Including such prompts could also risk introducing arbitrary assumptions about what these identities should look like. Furthermore, in many Latin American, Caribbean, and historical U.S. contexts, ‘mulatto’ was an official category in census data, legal documents, and colonial caste systems, influencing perceptions of race and identity for centuries. Consequently, while our conclusions are necessarily limited to the specific historical stereotypes associated with these terms, this focus ensures that our findings about the replication of pre-existing biases are clearly interpretable.  We thank the reviewer for raising this legitimate concern.
>
> **Weaknesses**
>
> W1: See response to Q1.
>
> W2: We appreciate the reviewer’s insightful comment regarding the narrow focus of comparing Europe and the US. As mentioned in the Authors’ Positionality statement, our perspective is shaped by our identities and lived experiences. Our focus on Europe stems from our understanding of its societal structures and ongoing racial discrimination. Expanding our analysis to other contexts, such as Africa or Asia, could lead to biased claims as we do not have the same knowledge and familiarity with these societies.
> That said, we believe the European example serves as an illustrative case rather than an exhaustive analysis. Different societies undoubtedly experience distinct forms of racism, many of which remain underexplored in current ML fairness research.
>
> W3: Regarding statistical robustness of the CFD-MR dataset (with 88 mixed-race individuals), we acknowledge that the sample size is small. However, this dataset is uniquely valuable because it is, to our knowledge, the only one that contains a self-reported mixed-race category, allowing us to analyze the complexity of mixed-race identity without imposing externally defined labels.
> Furthermore, our analysis of generative AI outputs, despite being based on this small sample, provides important insights into how these models tend to replicate pre-existing stereotypes. The scarcity of larger, self-reported mixed-race datasets reflects not only a gap in available data but also a critical shortfall in current data collection practices regarding certified, self-defined racial categories. This underscores the need for more robust and critically informed approaches to data collection to capture the complexities of mixed-race identity better.
>
> Based on your valuable feedback, we will clarify these points in the revised version of the paper.
>
> [1] CJEU, Judgment of 6 April 2017, Jyske Finans A/s v. Ligebehandlingsnævnet, C-668/15, ECLI:EU:C:2017:278.
>
> [2] European Commission: Directorate-General for Justice and Consumers, European network of legal experts in gender equality and non-discrimination, Chopin, I. and Germaine, C., A comparative analysis of non-discrimination law in Europe 2023 – The 27 EU Member States compared, Publications Office of the European Union, 2024.

---

### Decision · Program_Chairs · 2025-04-30

**Decision:**

Accept (spotlight poster)

**Comment:**

This paper addresses an unacknowledged question in ML fairness: whether treating race as a fixed input variable makes sense given the construct’s social, legal, and historical complexity. The writing is clear and precise, and the argument is supported with evidence drawn from law, sociology, and ML. Reviewer TbEj wrote, “The paper’s argument is clear, referencing a wide range of relevant works. It presents a logical flow, from critique of current practices to proposed solutions.” Reviewer 1qrB called the critique “a very fundamental problem in ML fairness.”

The biggest concern raised across reviews was practical implementation. Reviewer uy1p asked, “How would you operationalize ‘constitutive features’ in practice?” Reviewer 6ati noted that community engagement, while laudable, “would increase cost of data collection.” Reviewer TbEj also remarked that “there is a lack of real-world examples directly testing the proposed framework.” In response, the authors pointed to early efforts in hiring and healthcare domains and acknowledged the limitations of available datasets and framing those limitations as part of the motivation for the paper.

There was general enthusiasm for the contribution. Reviewer 6ati raised their score following the rebuttal, and reviewer 1qrB explicitly recommended acceptance.